# Perfusion and Ultrasonication Produce a Decellularized Porcine Whole-Ovary Scaffold with a Preserved Microarchitecture

**DOI:** 10.3390/cells12141864

**Published:** 2023-07-15

**Authors:** Gustavo Henrique Doná Rodrigues Almeida, Leandro Norberto da Silva-Júnior, Mariana Sversut Gibin, Henrique dos Santos, Bianca de Oliveira Horvath-Pereira, Leticia Beatriz Mazo Pinho, Mauro Luciano Baesso, Francielle Sato, Luzmarina Hernandes, Charles R. Long, Luciana Relly, Maria Angelica Miglino, Ana Claudia Oliveira Carreira

**Affiliations:** 1Department of Surgery, School of Veterinary Medicine and Animal Science, University of São Paulo, São Paulo 05508-270, Brazil; gustavohdra@usp.br (G.H.D.R.A.); silvajunior@usp.br (L.N.d.S.-J.); horvath@usp.br (B.d.O.H.-P.); mazoleticia@hotmail.com (L.B.M.P.); miglino@usp.br (M.A.M.); 2Department of Physics, State University of Maringá, Maringá 87020-900, Brazil; marigibin32@gmail.com (M.S.G.); rique.lovo@gmail.com (H.d.S.); mlbaesso@dfi.uem.br (M.L.B.); fsato@uem.br (F.S.); 3Department of Morphological Sciences, State University of Maringa, Maringá 87020-900, Brazil; lhernandes@uem.br; 4Department of Veterinary Physiology and Pharmacology, School of Veterinary Medicine & Biomedical Sciences, Texas A&M University, College Station, TX 77843, USA; clong@cvm.tamu.edu (C.R.L.); lurelly13@gmail.com (L.R.); 5Centre for Natural and Human Sciences, Federal University of ABC, Santo André, São Paulo 09210-580, Brazil

**Keywords:** ovaries, perfusion, decellularization, scaffold, reproduction, tissue engineering

## Abstract

The application of decellularized scaffolds for artificial tissue reconstruction has been an approach with great therapeutic potential in regenerative medicine. Recently, biomimetic ovarian tissue reconstruction was proposed to reestablish ovarian endocrine functions. Despite many decellularization methods proposed, there is no established protocol for whole ovaries by detergent perfusion that is able to preserve tissue macro and microstructure with higher efficiency. This generated biomaterial may have the potential to be applied for other purposes beyond reproduction and be translated to other areas in the tissue engineering field. Therefore, this study aimed to establish and standardize a protocol for porcine ovaries’ decellularization based on detergent perfusion and ultrasonication to obtain functional whole-ovary scaffolds. For that, porcine ovaries (*n* = 5) were perfused with detergents (0.5% SDS and 1% Triton X-100) and submitted to an ultrasonication bath to produce acellular scaffolds. The decellularization efficiency was evaluated by DAPI staining and total genomic DNA quantification. ECM morphological evaluation was performed by histological, immunohistochemistry, and ultrastructural analyses. ECM physico-chemical composition was evaluated using FTIR and Raman spectroscopy. A cytocompatibility and cell adhesion assay using murine fibroblasts was performed. Results showed that the proposed method was able to remove cellular components efficiently. There was no significant ECM component loss in relation to native tissue, and the scaffolds were cytocompatible and allowed cell attachment. In conclusion, the proposed decellularization protocol produced whole-ovaries scaffolds with preserved ECM composition and great potential for application in tissue engineering.

## 1. Introduction

Tissue engineering is an emerging field in biotechnology that develops innovative solutions to repair, regenerate, or replace injured tissues and organs [1]. Great advances were accomplished for vital organs such as the liver, heart, and kidneys [2,3,4] and for support and locomotion tissues such as bone, cartilage, and skeletal muscle to reverse degeneration in tissues with low regenerative capacity [5,6]. More recently, it was proposed to apply these principles to reproduction medicine [7].

Female reproductive tissues have singularities that distinguish them from the other tissues. They are highly responsive to sex hormones, which cause morphophysiological alterations, highlighting great periodic plasticity [8,9]. This plasticity makes the development of biomimetic reproductive tissues more challenging, as the structure that supports the system must be adaptable to hormonal changes and mimetize more accurately the molecular microenvironment similar to what occurs in vivo [7,10].

The ovary has a unique microarchitecture that contributes directly to ovarian follicle maturation and maintenance, which provides a suitable microenvironment for oocyte survival [11,12]. Ovarian tissue impairment may occur through ovarian pathologies such as polycystic ovary syndrome (POS), systemic metabolic disorders such as diabetes and hypertension, and exogenous agents such as chemotherapy that cause ovarian tissue degeneration [13,14]. Tissue and oocyte cryopreservation therapies aim to preserve female fertility by storing the germ cells in suitable conditions to remain viable [15,16,17]. Although effective, such approaches are limited due to methodological drawbacks such as high execution costs, the application of cytotoxic cryoprotection, and oocyte degeneration associated with freezing/thawing or vitrification processes [18,19,20,21,22].

Studies have attempted to develop biomimetic ovarian tissues capable of supporting in vitro ovarian follicle culture or that were transplantable, reestablishing ovarian functions in infertility models [23,24,25,26,27]. Several biomaterials, both natural and synthetic, were used to recreate the structural and functional support for the biomimetic ovarian tissue; however, despite the promising results, such materials are not able to recapitulate with high accuracy the extracellular matrix (ECM) biological properties to reconstruct the ovarian microenvironment [9,28,29].

To overcome these limitations, it was proposed to use decellularized scaffolds, which present native tissue 3D structure and ovarian ECM components, which are essential for folliculogenesis, follicular maturation, and oocyte survival [30,31,32,33,34]. For efficient decellularization, a balance between the aggressiveness of cellular removal and ECM preservation is required [35,36]. For this, physical, chemical, and enzymatic methods are used, isolated or in association, to obtain functional acellular scaffolds [35,36].

Ovarian decellularization has been reported for murine, bovine, swine, and human ovaries [32,37,38]. However, in all cases, only small ovarian fragments were decellularized for application in functional assays [32]. Only two studies from the same group reported whole ovaries decellularization [38,39]. Pennarosa et al. (2020) described porcine whole ovaries’ decellularization using a combination of anionic and non-anionic detergents [38]. However, a methodological limitation of all studies to date has been the immersion of tissues in decellularizing solutions and the use of orbital agitation to promote the removal of cellular components [35,40]. This method, although effective, may compromise tissue microstructures and lead to the loss of ECM components due to mechanical action and long exposition to the reagents [35,40].

An alternative to this method is the perfusion of solvents through the vasculature of the whole organ to diffuse the solutions throughout the tissue in a more controlled way that preserves the structures [40,41]. The ovary in particular is covered by a fibrous tunic that makes it difficult for solutions to penetrate deeper compartments, which justifies the use of the perfusion method [42]. Another complementary resource to decellularization chemical methods is physical methodologies such as the use of ultrasonication. Ultrasonication can rupture cell membranes and remove cellular debris that may remain intertwined in the extracellular matrix [36,43]. This combination can decellularize the ovary more efficiently and preserve ECM components, generating an entire acellular ovarian scaffold that maintains the complex micro-architecture and has a large potential for application in reproductive medicine and other tissue engineering applications [9,30,44].

Therefore, this study aimed to establish a new and efficient protocol for the entire porcine ovaries decellularization through detergents perfusion and ultrasonication application to generate an acellular scaffold with the three-dimensional microstructure and ECM components preserved that allows cell culture and survival.

## 2. Materials and Methods

### 2.1. Ovarian Samples Acquisition

Samples were obtained from the slaughterhouse of the Faculty of Animal Science and Food Engineering of the University of São Paulo, Pirassununga campus, Brazil. The ovaries (*n* = 5 per group) were collected from prepubertal sows at approximately 6 months of age. Immediately after the collection, the samples were stored on ice and transported to the laboratory. Adjacent ligaments and uterine remnants were removed, and ovarian major vessels were dissected. This research was conducted in compliance with the institutional ethics committee regulations of the University of São Paulo (CEUAx protocol no. 8756210222). The methodological pathway is summarized in Figure 1. 

### 2.2. Whole-Ovaries Decellularization

The ovaries were cannulated through the ovarian artery and perfused with the assistance of an ORCA Bioreactor™ (Harvard Apparatus, Holliston, MA, USA) perfusion bomb. The perfusion sequence took place in three stages: (1) initial washing—2 h of deionized water (dH_2_O) and 2 h of sodium phosphate buffer (PBS) 1x at a rate of 1 mL/min; (2) decellularization—sodium dodecyl phosphate (SDS) 0.5% for 48 h and Triton X-100 1% for 5 h at a rate of 1 mL/min; (3) final washing—the ovaries follicles were ruptured with hypodermic needles to facilitate cell debris washing, then perfused for 24 h with deionized water and 24 h with sodium phosphate buffer (PBS) 1x at a rate of 3 mL/min while they underwent an ultrasonic bath for 15 min. After that, the ovarian follicles were ruptured with injection needles to facilitate the cell debris washing, and then deionized water was perfused at a rate of 3 mL/min in the ultrasonic bath (SOLIDSTEEL, Piracicaba, Brazil), 3 cycles of 5 min. Ovarian scaffolds were stored in PBS 1x at 4 °C for further analysis.

### 2.3. 4,6-Diamidino-2-Fenilindole (DAPI) Staining

DAPI fluorescent staining was used to verify nuclei or DNA fragments’ presence after the decellularization process. Sample fragments were frozen in Tissue Plus O.C.T. compound optimal cutting temperature (Fisher Health Care, Houston, TX, USA) and microsectioned using a cryostat (CM1860 model, Leica Biosystems, Baden-Wurttemberg, Germany). The slides were stained by DAPI solution (1:10,000) at room temperature without light for 10 min and then washed with PBS 1x for subsequent analysis using fluorescent microscopy (Nikon ECLIPSE 80I, CADI FMVZ-USP, Tokyo, Japan).

### 2.4. Histological Analysis

Histological evaluation was used to verify the decellularization protocol’s efficiency. Native and decellularized ovarian fragments from cortical and medullary regions were fixed in 4% buffered paraformaldehyde for 48 h, dehydrated using increasing concentrations of alcohol (70, 80, 90, and 100%), diaphanized in xylol, and embedded in paraffin. Sections of 5 µm (No. RM2265; Leica) were stained with hematoxylin and eosin (HE) to evaluate nuclei presence and ECM general condition; Masson’s trichrome for total collagen content detection; Picrosirius red to distinguish different stages of collagen maturation; Alcian blue (pH = 2.5) to evaluate GAGs general content; and Weigert’s fuchsin-resorcin to evidence elastic fibers. Slides were photographed and analyzed using a light microscope (Nikon ECLIPSE 80I, CADI FMVZ-USP).

### 2.5. Scanning Electronic Microscopy (SEM)

Ovarian samples were washed four times with deionized water in the ultrasonic bath, then in Karnovsky solution (2.5% glutaraldehyde and 4% paraformaldehyde in a buffered 0.1 M sodium cacodylate) for 48 h and dehydrated in increasing alcohol concentrations for 5 min each. After that, the fragments were dried in a supercritical point device (LEICA EM CPD 300^®^) and then sputter coated with gold (EMITECH K550^®^, Quorum Technologies, United Kingdom). Finally, the samples were photographed under a scanning electron microscope (LEO 435 VP^®^, Oberkochen, Germany).

### 2.6. Immunohistochemistry Analysis

The sections were rehydrated in citrate buffer in the microwave for antigen retrieval. The endogenous peroxidase blockage was performed with 3% hydrogen peroxide in distilled water for 30 min in the dark. After that, 2% bovine serum albumin (BSA) in PBS was used for non-specific protein interaction blockage. The primary antibodies used were: anti-collagen I (#PA5-29569, 1:250, Invitrogen, Carlsbad, CA, USA), anti-collagen III (#PA1-28870, 1:250; Invitrogen), anti-fibronectin (#Ab2413, 1:100, Abcam, Cambridge, UK), anti-laminin subunit α2 (#PA1-16730, 1:200; Invitrogen), anti-elastin (#Ab9519, 1:100, Abcam), anti-hyaluronic acid (#c41975, 1:100, LS Bio, Seattle, WA, USA), and the secondary antibodies were IgG anti-mouse/anti-rabbit (#K800; Dako, CA, United States). The incubation occurred overnight in a wet chamber at 4 °C. The reaction was detected by Dako Advance HRP (#K6068; Dako) and developed with DAB (#k3468; Dako), according to the manufacturer’s instructions. Slides were photographed and analyzed using a light microscope (Nikon ECLIPSE 80I, CADI FMVZ-USP).

### 2.7. Genomic DNA Quantification

For DNA quantification, the QIAamp^®^ DNA Mini Kit (Qiagen, Hilden, Germany) was used to extract the genetic material from native and decellularized samples according to the manufacturer’s specifications. The fragments were digested overnight at 56 °C by the action of Proteinase K stabilized by the kit’s lysis buffer. The fragments were purified and analyzed by spectrophotometry at 260 nm (Nanodrop, Thermo Scientific, Waltham, MA, USA).

### 2.8. Fourier Transform Infrared Spectroscopy (FTIR) Analysis

Fourier transform infrared spectroscopy (FTIR) was used to evaluate the ECM molecular composition of native and decellularized samples. The analysis was performed in a Bruker Vertex 70v FTIR spectrometer (Bruker Optik GmbH, Ettilingen, Germany) with an attenuated total reflectance (ATR) accessory. The ovary samples (*n* = 5) were sectioned into three regions: cortical, corticalmedullary (intermediate), and medullary regions. This compartmentalization was conducted to evaluate possible differences in the ECM molecular profile of each region. The spectrum of each ovary region, between 4000 and 400 cm^−1^, is an average of three measurements with 128 scans and 4 cm^−1^ of spectral resolution. All measurements were performed at room temperature, and the spectra were vector normalized using OPUS software 8.7 SP2.

### 2.9. Raman Spectroscopy Analysis

Complementary to the FTIR technique, Raman spectroscopy was also used to detect physical-chemical alterations in native and decellularized ovarian ECMs. The samples (n = 5) were sectioned into three regions as mentioned. The Raman spectra were obtained using a Senterra Confocal Raman microscope (Bruker Optik GmbH, Ettilingen, Germany) equipped with an objective lens (20× magnification) focusing the excitation laser (785 nm, 100 mW) on the sample. The spectrum of each ovary region, from 1750 to 400 cm^−1^, is an average of four measurements with 30 scans, 4 cm^−1^ of spectral resolution, and 10 s of detector integration time. All measurements were performed at room temperature and the spectra were vector normalized using OPUS software 8.7 SP2.

### 2.10. Scaffolds Sterilization

After decellularization and washing, the whole-ovary scaffolds were perfused for 10 min with 70% alcohol, then the scaffolds were fragmented and immersed in progressive and regressive alcohol concentrations (70%, 80%, 90%, 100%, 90%, 80%, and 70%) for 5 min each in the laminar flow hood. Finalizing the alcohol cycle, the fragments were washed five times with PBS 1x with 2% antibiotics (Penicillin-Streptomycin 10,000 µg/mL, LGC Biotecnologia, Cotia, Brazil) to remove the alcohol. Then, the scaffolds were exposed to UV light for 5 min and posteriorly sealed for analysis. To evaluate the scaffold sterility, the fragments were immersed in culture medium and placed into the cell culture incubator for 72 h in α-MEM medium (Sigma, St. Louis, MO, USA) supplemented with 10% bovine serum (Invitrogen Co. Ltd., Carlsbad, CA, USA).

### 2.11. Cytocompatibility Assay

After the sterilization process, 5.0 × 10^4^ 3T3 cells were seeded on scaffold fragments immersed in supplemented α-MEM medium for 7 days at the same conditions previously described to evaluate cell adhesion and viability. The seeded scaffolds were collected and processed for histological analysis, DAPI staining, and SEM analysis.

### 2.12. Statistical Analysis

Shapiro—Wilk normality tests were conducted to confirm the normal distribution of the data. DNA content was analyzed by Mann-Whitney test. Statistical significance was considered at *p* < 0.05. Data were presented as the mean ± SD. PCA analysis was performed on FTIR and Raman data. A one-way ANOVA was performed to compare the means of quantitative FTIR-ATR and Raman quantitative data. Statistical significance was considered when *p* < 0.05. A Turkey post hoc test was applied to compare native and decellularized samples. The data were analyzed with GraphPad Prism 7.0 (GraphPad Software, Inc., San Diego, CA, USA).

## 3. Results

### 3.1. Whole-Ovary Perfusion Decellularization

Detergent perfusion involves accurate and stable cannulation of the chosen vessel to allow efficient solution diffusion throughout the organ [45]. Ovarian blood circulation is simple, consisting of one major artery and a vein. The chosen vessel for the procedure was the ovarian artery, which was dissected and cannulated using a 21G needle that was tightly tied and stabilized with a nylon thread (Figure 2A). At each stage of the perfusion process (dH_2_O, PBS, SDS, Triton X-100, and final washing), there was a progressive whitening of the ovary until the final stage (Figure 2D), which got completely white and translucent after the decellularization process (Figure 2B). The macroscopic ovaries appearance was used as a primary parameter for the decellularization. Preliminary histological analyses revealed that, despite the decellularization process having occurred with the mentioned solution combination, a cellular debris accumulation was noted inside the intact ovarian follicles, quite possibly because they are an isolated structure surrounded by a dense follicular basal membrane. Due to this observation, follicles were manually ruptured to increase the perfusion rate during the last washes from 1 to 3 mL/min and put into an ultrasonic bath to remove remaining cellular debris (Figure 2C). H&E and DAPI staining histological results showed nuclei absence (Figure 2H,I,L,M) and ovarian ECM general structure preservation (Figure 2H,I). These results were supported by DNA quantification, which revealed that the protocol reduced the DNA content by 97% (Figure 2E).

### 3.2. Structural and Ultrastructural Characterization

Once the cellular removal efficiency of the protocol was attested, the morphological (Figure 3), immunohistochemical (Figure 4), and ultrastructural (Figure 5) characterization of the decellularized material was performed to verify whether the main ovarian ECM components were preserved. Through Masson’s trichrome stain, we observed that the total collagen content, stained in blue, was preserved without apparent collagen fiber bundles disarrangement in decellularized samples (Figure 3C,D) in relation to the native tissue (Figure 3A,B). Masson’s trichrome staining also corroborated previous findings regarding cellular removal because decellularized samples did not show dark or black stained nuclei, which were evident in native tissue.

Collagen fibers in ovarian tissue do not present the same arrangement in all its compartments, with different maturation levels along tissue composition [46]. To observe these collagen fiber arrangement stages in the samples, Picrosirius red staining was used, which stains the total collagen content in red (Figure 3E–H); however, when exposed to polarized light, it distinguishes mature collagen (reddish and yellowish tones) from immature collagen (greenish tones) according to birefringence. The results showed that the decellularized samples (Figure 3I,J) presented the same preservation pattern for both types of collagen, similar to native ovary samples (Figure 3K).

The glycosaminoglycans (GAGs) content was demonstrated by Alcian blue staining, which stains the regions containing these components in bluish tones (Figure 3M,P). The results showed that decellularized tissue maintained the glycosaminoglycans after the decellularization process (Figure 3O,P) in comparison to the native tissue (Figure 3M,N).

The elastic fibers were highlighted by Weigert’s fuchsin-resorcin staining, which stains the elastic components in dark tones (Figure 3Q–T). The findings showed that the decellularized samples’ elastic components (Figure 3S,T) were preserved regarding the native ovarian tissue (Figure 3Q,R).

Immunohistochemical analyses were performed to confirm the presence of the main ECM components and if they remained after the decellularization process (Figure 5). Type I and III collagen, elastin, laminin and fibronectin, glycoproteins, and hyaluronic acid, a non-sulphated glycosaminoglycan, which comprise the most abundant ECM components, were immunostained in native and decellularized samples. Type I and III collagens and elastin were the most prominent proteins in the ovarian stroma, with their expression proving a similar pattern in native and decellularized ovaries (Figure 5A–F). Laminin expression was limited to blood vessels and the ovarian follicle compartment, with a similar expression in both groups (Figure 5G,H). Fibronectin immunostaining was also circumscribed to the follicle compartment, with the expression slightly enhanced in decellularized samples, probably due to the absence of cells (Figure 5I,J). Regarding hyaluronic acid, the expression was detected in several portions of the ovary, being very similar in both groups (Figure 4K,L).

After histological and immunohistochemical analysis, an ultrastructural analysis of native (Figure 5A,C,E,G) and decellularized (Figure 5B,D,F,H) samples was performed, and a comparison was drawn to verify the decellularization protocol’s efficiency and characterize the scaffolds three-dimensional structure. The lowest magnification revealed that even after the decellularization process, the ovarian follicle structure and stroma organization were preserved, as was the fibrous tunica albuginea that covers the whole organ (Figure 5B). While the native samples showed filled follicles and stromal tissue covered with cells (Figure 5C), the scaffolds were acellular, with the fibrous arrangement preserved and structured (Figure 5D). At the highest magnification, it was observed that the fibers in the native samples were spaced due to the interspersed cells (Figure 5E,G), whereas the decellularized samples showed closer and more compact fiber bundles due to the cell absence (Figure 5F,H). Scanning electron microscopy also showed that both the densest and most resistant collagen fibers as well as the thinnest and most delicate ones were preserved, corroborating the histological findings and demonstrating that the produced scaffolds maintained their three-dimensional structure.

### 3.3. ECM Composition Characterization by Spectroscopic Analyses

Both FTIR-ATR and Raman spectroscopy were used to evaluate physico-chemical alterations in the ECM from native and decellularized samples. These vibrational spectroscopies performed together promote a better understanding of the chemical characteristics of the samples since each one provides a different response of radiation–matter interaction, complementing each other. For a preliminary evaluation, the ovaries were sectioned into three portions: cortical, corticomedullary, and medullary regions. This classification was made to verify if there was a spectroscopic distinction between the regions regarding the ECM composition. The statistical method used to analyze the found spectra was principal component analysis (PCA) for both sample groups. The FTIR-ATR spectra of native and decellularized samples (Figure 6A) exhibited bands associated with amide A, characterized by NH stretches coupled to hydrogen bonds, while amide B is signed by asymmetric CH_2_ stretches and may exhibit overlaps with the amide A band. Amides A and B can be found in the spectral range of 3440 to 3300 cm^−1^ and 2990 to 2830 cm^−1^, respectively [47,48]. The band centered at 1640 cm^−1^ is characteristic of amide I, mainly attributed to carbonyl C=O stretching along the protein chain [47,49]. Amide II and III can be found with centers at approximately 1550 cm^−1^ and 1240 cm^−1^, respectively. These are associated with NH and CH_2_ molecules bending, as well as C–N bonds stretching [50]. Furthermore, amide III is related to C–O stretching [47,48]. Finally, the range from 1140 to 985 cm^−1^ can be related to C–O–C and C–OH vibrations [49]. Figure 5A represents the spectra mean for each one of the regions. Figure 6B,C represent the PCA scores for native and decellularized samples, respectively. This result indicates that there is no spectral difference between the ovarian-sectioned regions, showing that the regions present contributions in similar bands.

Principal components analysis (PCA) explored differences in the ovary regions spectra. Figure 6B,C present PCA scores for native and decellularized samples, respectively, showing that there is no spectral difference between the ovarian-sectioned regions. Therefore, the average spectrum among the ovary regions was carried out in each native and decellularized group, as shown in Figure 7A.

Figure 7B shows that the PCA score plot groups the spectra into PC1 > 0 for decellularized and PC1 < 0 for native, with PC1 containing 89.8% of the total variance among data sets while PC2 represents only 6.4%. This result showed that there is a spectral difference between native and decellularized samples in terms of the PC1 component, confirming that both groups can be measured independently. PC1 loading showed differences over all spectral ranges, with the amide I and II regions being responsible for highlighting and defining the differentiation between the groups (Figure 7C).

Amide regions can be associated with organs with high collagen content, with amide I serving as a spectral marker for protein secondary structure. Amide II can be related to protein hydration but also can indicate collagen self-assembly, which consists of a spontaneous aggregation of several scales of collagen molecules to form staggered longitudinal matrices [47,51]. The triple helical structure of collagen can be evaluated by means of the ratio of amide III to 1450 cm^−1^ (CH_2_ deformation) [47]. C–O–C and C–OH bonds are associated with proteoglycan content, which is composed mostly of sulfated glycosaminoglycans (GAGs) bound covalently to a protein core [47,52]. Based on these spectra screening results, to evaluate the content of compounds and protein structures before and after the decellularization process, band areas of amides I, II, and proteoglycan were obtained by integration, and the ratio of amide III: 1450 cm^−1^ was calculated as shown in Figure 8. There was no statistically significant difference in the proteoglycan content and collagen’s triple helical structures between the native and decellularized samples. Regarding the collagen content, decellularized samples presented a percentual variation of 11% for amide I and 27% for amide II. This variation is not related to an increase in collagen content, but due to the cell removal, collagen fibers probably self-assembled, enhancing the spectral intensity of collagen molecular components.

In addition to FTIR results, Raman spectroscopy analysis was also applied to evaluate ECM physico-chemical alterations in both native and decellularized tissues. Raman spectra before and after the decellularization process (Figure 9A) exhibited bands associated with the amide I band at 1660 cm^−1^, the amide III band between 1288 and 1218 cm^−1^ [52,53,54,55,56,57], and the vibrations of CH_2_ and CH_3_ molecules [55] centered on a band at 1450 cm^−1^. S=O symmetrical stretching at 1062 cm^−1^ can be attributed to GAG molecules and the ring stretching vibration mode of phenylalanine at 1003 cm^−1^ [53,54,57,58,59,60]. The overlapping bands at 856 and 875 cm^−1^ can be associated with the presence of C–C stretches present in the proline and hydroxyproline ring residues [54,55,56,57,58]. The elastin can be characterized at 725 cm^−1^ [55,58]. Raman band areas of amide I, amide II, phenylalanine, proline, GAGs, and elastin were calculated by integration to compare their contributions before and after the decellularization process, as shown in Figure 9B. It is consistent with the FTIR-ATR results that ECM compounds did not present a statistically significant variation. Therefore, it is possible to infer that no variations were detected in the extracellular matrix components after the decellularization protocol.

### 3.4. Cytocompatibility Evaluation

Once the ECM composition of the scaffolds was characterized, an assay was conducted to evaluate the scaffolds’ cytocompatibility. To evaluate cell interaction with the scaffolds, 3T3 fibroblasts were seeded and cultured for 7 days in order to evaluate if the cells were able to survive, anchor, and migrate through the scaffold (Figure 10). Using H&E staining (Figure 10G–I) and DAPI staining (Figure 10J–L), it is possible to observe that the cells after the culture period remained on the scaffold’s surface. SEM photomicrographs (Figure 10A–F) showed fibroblasts attached to the ECM acellular scaffolds through cell membrane projections. Cells are visibly anchored on the scaffolds, which allowed for their interaction along the surface. These results demonstrated that the generated scaffolds allow cell attachment, probably related to the preservation of adhesive glycoproteins that mediate cell adhesion and, thus, interaction along the scaffold.

## 4. Discussion

Decellularized ovarian scaffolds have already been described as promising platforms for cell culture and excellent sources of biological matrixes for hydrogel production, which may be combined with other biomaterials and applied for in vitro oocyte maturation and maintenance [30,61,62,63,64,65]. However, in most of these studies, the decellularization processes are based on ovarian fragment immersion in a combination of solutions to remove cellular components [66]. Such methodology, in addition to disrupting the ovarian macrostructure and dissociating the distinct ovary, involves prolonged exposure of the tissue to detergent and the mechanical action of orbital agitation, which is able to degrade the ECM components and the scaffold microstructures, producing materials of lower biological quality [67,68,69,70].

Pre-pubertal porcine ovaries were chosen for this study because, as these animals were not into the reproductive cycle yet, the samples would be less variable and demonstrate similar structural and physiological properties as it is known that hormonal stimulation promotes drastic changes in ovarian tissue, and therefore in the ECM [23,71,72,73,74,75]. A second reason is the great availability of this material; these organs are discarded by slaughterhouses, which reinforces the initiative to reuse these tissues for biotechnological purposes [76,77]. Another reason is the translational potential of porcine decellularized ovaries to human clinical trials; despite the physiological differences and the fact that there is no ideal animal model to mimetize the human ovary, porcine follicular waves have a greater level of similarity with humans than other species such as bovine, ovine, and even non-human primates [71]. A recent proteomic study analyzing decellularized porcine showed that ovaries have an enriched matrisome with great potential to be applied as a biomaterial for tissue engineering [71].

A new decellularization protocol was presented to generate a biological acellular scaffold of the whole porcine ovary based on the perfusion of 0.5% SDS and 1% Triton X-100 detergents and ultrasonication. The detergents perfusion through the vascular system with a controlled flow rate allows the solutions to penetrate easily into the organ parenchyma, infusing into regions of difficult access with little contact surface [69]. This technique also allowed for macro- and microstructural integrity preservation [69].

Only three studies proposed decellularization methods for larger ovarian tissue portions or for the whole organ [33,38,78]. Hassanpour et al. [35] decellularized human hemi ovaries using DNAse I and 1% sodium lauryl ester sulfate (SLES), generating non-cytotoxic scaffolds with preserved ECM structure and composition [33]. The main limitations of the study are the long-term process that takes 30 days to be completed, which makes the protocol difficult to implement since it requires a large number of detergents for decellularization, and the delay to generate an acellular scaffold, which makes its application problematic [33]. Another drawback is the application of DNAse I, which is a high-cost reagent and makes the protocol more expensive.

Pennarosa et al. [38,39] were the first group to propose the whole ovary decellularization. The protocol is based on the immersion and orbital agitation of porcine ovaries in 0.5% SDS for 3 h, followed by 1% Triton X-100 for 9 h, and then in 2% SDC for 12 h, obtaining decellularized whole ovarian scaffolds [38,39]. Although the results were satisfactory, the data differed from ours using the perfusion protocol. The perfused porcine ovaries remained exposed for a longer period in the detergents, but the concentrations, exposure period, and applied procedures did not impair the scaffold’s three-dimensional structure, as shown in SEM images, as well as the ovarian ECM composition.

Different from the currently used methods for ovarian decellularization, the described protocol included ultrasonication, a physical method that provided additional stimulation to assist in cell rupture and debris removal without degradation of the ECM [79,80]. Similar systems were used to decellularize porcine kidneys, in which perfused decellularizing solutions were concomitantly applied to ultrasonication to obtain acellular porcine renal scaffolds [81,82]. These studies showed that the sonication power and flow rate affect the exposure period to the detergent and the total decellularization period [81,82]. These findings support our use of ultrasonication as a complementary method to chemical decellularization; nevertheless, there must be a balance in the application of this method, because tissue exposure to high sonication power can cause damage to tissue microstructure and decrease the detergent efficiency in solubilizing cell membrane lipids [83].

The acquisition and maintenance of three-dimensional structures of several organs and tissues have been one of the main tissue-engineering goals [84,85]. Several studies have already attested that the biological response occurs much more reliably in complex 3D systems analogous to those found in vivo compared to 2D systems [86]. Tissue structure is closely related to the spatial and functional cellular organization in the tissue microenvironment [87]. In this context, the extracellular matrix acts as a cellular signaling factor, which provides molecular coordinates for cell anchorage and establishment [88].

Regarding the ovary, a compartmentalized organ whose portions have distinct and defined compositions and functions, the structure is essential for follicular maturation and maintenance [42]. Several studies have already demonstrated that there is a difference between the mechanical strengths exercised by ovarian ECM in the cortical and medullary regions [89]. This tensile distinction acts directly on follicular differentiation and development processes; ECM stress fiber elasticity promotes mechanical signals, which influence cytoskeleton proteins through mechanotransduction, modulating their organization [90]. This modulation also influences intracellular protein activation, which activates transcription factors and regulates specific target gene expression [91]. The mechanical influence is able to regulate the process of oocyte dormancy, with mechanical stress loosening only at the time of oocyte activation, which is expelled at ovulation [89].

Histological and immunohistochemical findings suggest that ovarian ECM components were preserved in comparison to the native tissue. However, instead of performing image quantification that may contain errors, be biased, or not represent the entire tissue scenario [92], and to provide more robustness to the data and more accurately quantify the percentage of preservation, spectroscopic analyses by FTIR and Raman, which are more sensitive techniques, were performed to detect physicochemical alterations in ECM molecules [53,58]. One relevant result of our study was the spectroscopic similarity between the distinct ovarian regions because although ECM components have a different organization in cortical and medullary regions, in terms of molecular composition, native and decellularized prepubertal porcine ovaries were similar. This result confirms that the perfusion protocol was able to decellularize the entire ovary equally, with no significant loss of ECM components in any of the analyzed regions.

A suitable biomaterial for biotechnological applications can support and maintain viable cells and allow cellular adhesion, migration, and proliferation [89]. Decellularized ovarian ECM has the potential to support not only native ovarian and follicular cells but other cell types that are not found physiologically, attesting that these scaffolds may be applied as cell culture platforms for a great number of experiments and purposes or as a source of biomaterial that may be converted into hydrogels or associated with other compounds to develop new sorts of biopolymers [31,65,66,90]. Our results demonstrated that the generated scaffolds allowed cell adhesion of murine fibroblasts, highlighting that these biomaterials afford conditions for cell survival and adhesion. The interaction between porcine ovarian ECM and murine fibroblasts also demonstrated that the biomaterial could support non-ovarian native cells, revealing their potential to be used in other tissue engineering applications and not restricting the ovarian ECM to ovarian reconstruction or only to reproductive medicine purposes. These data altogether provide enough evidence that such scaffolds can be applied for ovarian microenvironment reconstruction and may be used as a cell culture platform to develop in vitro models for ovarian tissue, drug screening assays, germ cell differentiation, and even be translated to other tissues, opening a new world of possibilities for ovarian ECM-derived biomaterials.

## 5. Conclusions

In this study, we established an efficient and reproducible protocol for porcine, whole ovary decellularization combining perfusion and ultrasonication. This protocol resulted in a whole-organ scaffold with the microarchitecture preserved, which indicates potential application in reproductive tissue engineering. The generated scaffolds were acellular, with the ECM components preserved, and non-cytotoxic, which allowed cell viability and attachment. The technique produced a viable and suitable biomaterial to be applied as a platform for ovarian microenvironment reconstruction and to support cell culture and application in in vitro reproduction biotechnologies.

## Figures and Tables

**Figure 1 cells-12-01864-f001:**
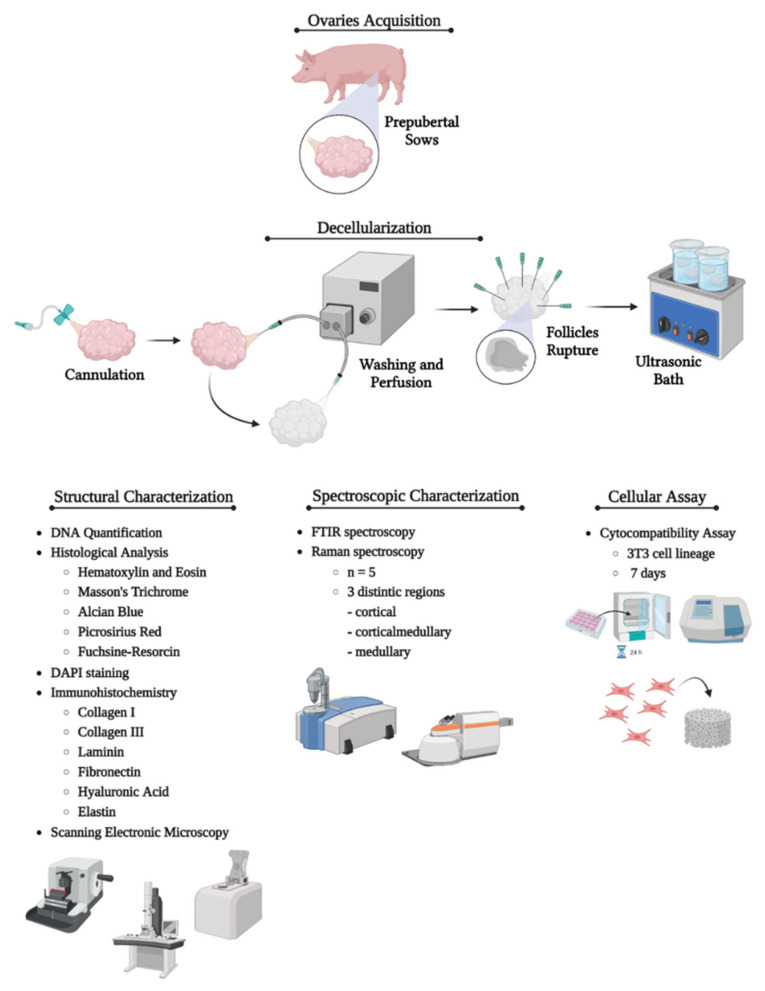
Representative scheme showing the study’s methodological pathway. The first step consisted of porcine ovaries acquisition. Then, the ovarian artery was cannulated, and the ovaries were perfused with deionized water, PBS, detergents, and ultrasonication to accomplish the decellularization. Once decellularized, the whole-organ scaffolds underwent structural, spectroscopic, and cellular assays to characterize the material composition, decellularization protocol efficiency, and the scaffold’s biological properties related to cell anchorage, survival, and proliferation. Created with BioRender.com.

**Figure 2 cells-12-01864-f002:**
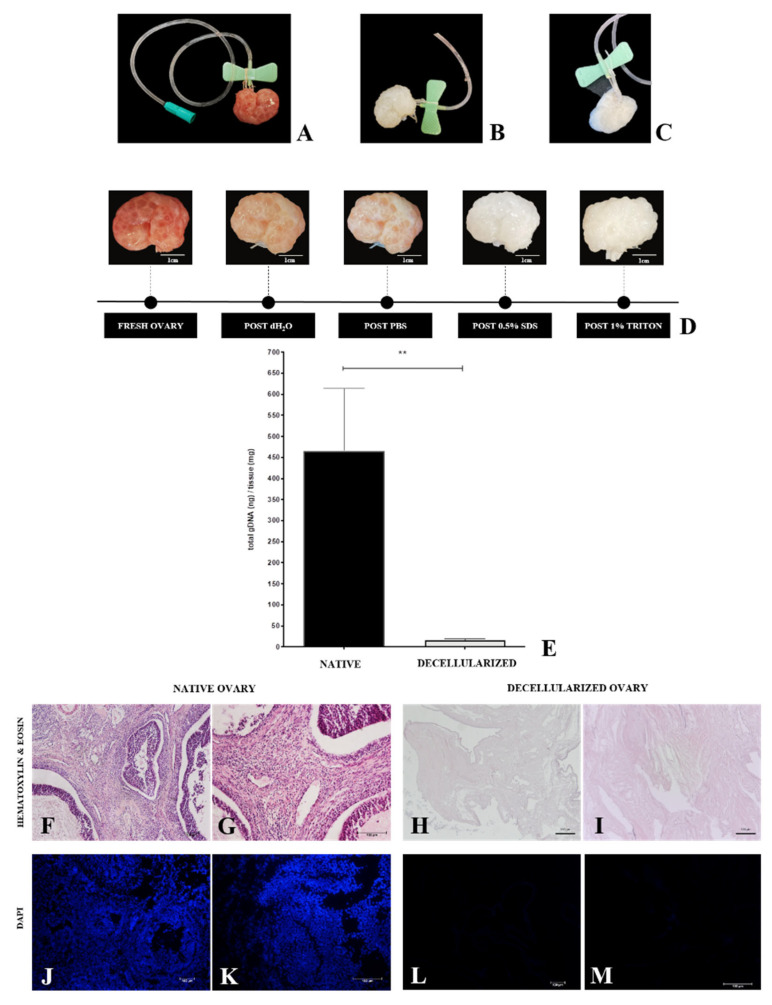
Whole-ovary perfusion decellularization efficiency evaluation. Cannulated ovary through the ovarian artery (**A**). Decellularized whole ovary (**B**). Decellularized whole ovary with ruptured follicles (**C**). Macroscopic aspect of perfusion ovaries, highlighting the progressive whitening (**D**). DNA quantification (**E**) and histological evaluation (H&E and DAPI staining) of samples undergone perfusion decellularization (**H**,**I**,**L**,**M**) compared to native samples (**F**,**G**,**J**,**K**). ** *p* < 0.05 compared to the native tissue.

**Figure 3 cells-12-01864-f003:**
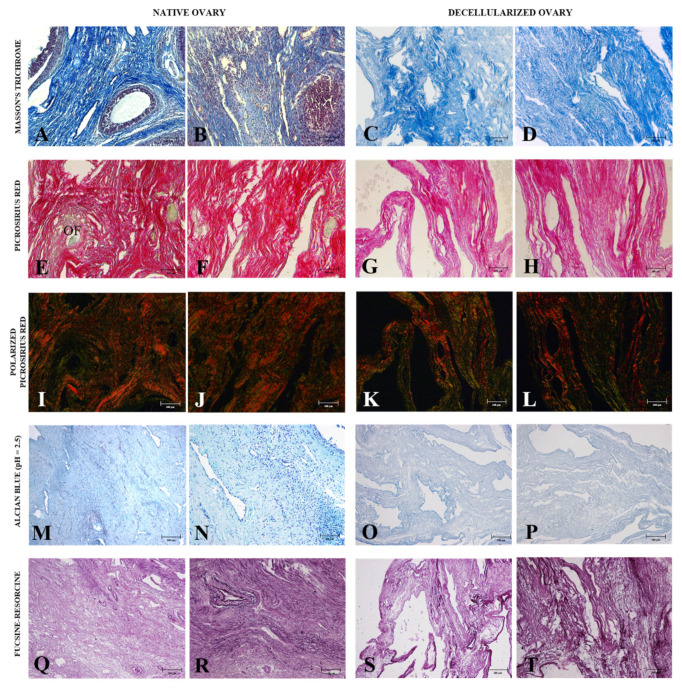
Morphological analysis of ovarian ECM from native and decellularized ovaries. Masson’s trichrome staining highlights collagen content (blue) in native (**A**,**B**) and decellularized samples (**C**,**D**). Non-polarized Picrosirius red staining confirms the collagen fiber retention in native (**E**,**F**) and decellularized tissues (**G**,**H**). Polarized Picrosirius red staining differs collagen fibers in different maturation stages in native (**I**,**J**) and decellularized samples (**K**,**L**). Mature collagen is stained in reddish and yellowish tones and immature collagen fibers are stained in greenish tones. Alcian blue (pH = 2.5) staining highlights GAGs content in native (**M**,**N**) and decellularized tissues (**O**,**P**). Weigert’s fuchsin-resorcin staining shows the elastic fibers distribution in native (**Q**,**R**) and decellularized samples (**S**,**T**). OF (ovarian follicle).

**Figure 4 cells-12-01864-f004:**
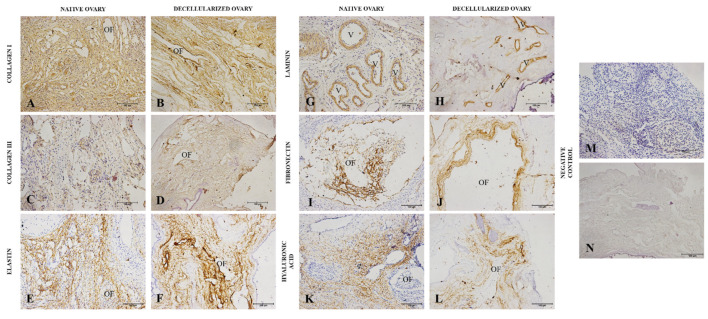
Immunohistochemistry of main ovarian ECM macromolecules from native and decellularized ovaries. IHC staining highlights the expression of type I collagen (**A**,**B**), type III collagen (**C**,**D**), elastin (**E**,**F**), laminin (**G**,**H**), fibronectin (**I**,**J**), and hyaluronic acid (**K**,**L**) both in native and decellularized samples. Negative control (**M**,**N**). OF (ovarian follicle), V (vessel).

**Figure 5 cells-12-01864-f005:**
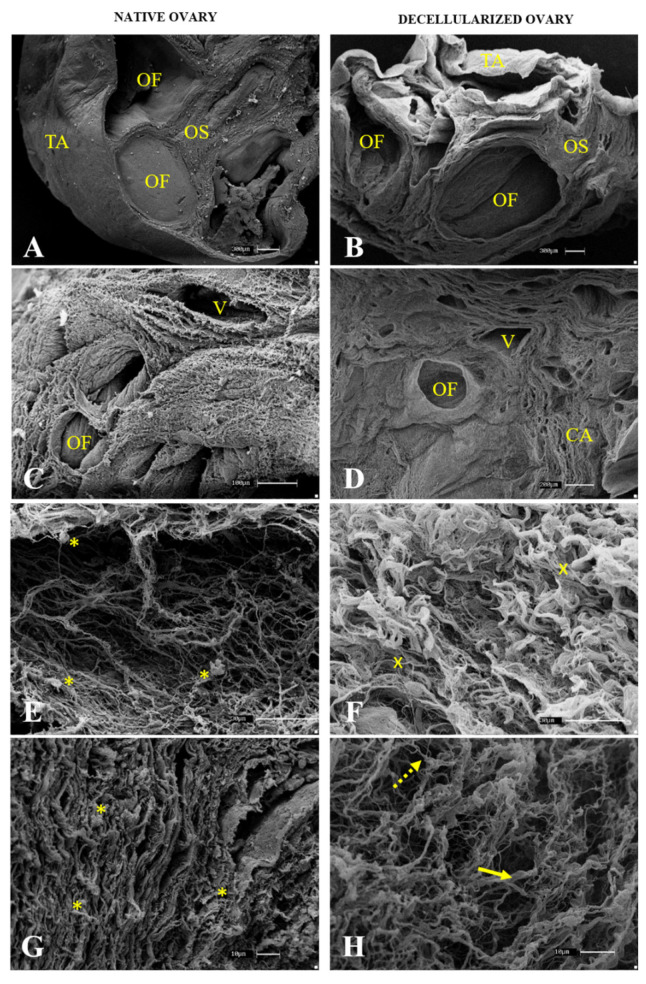
Representative scanning electron micrographs of ovarian native (**A**,**C**,**E**,**G**) and decellularized tissue (**B**,**D**,**F**,**H**). OF (ovarian follicle), TA (tunica albuginea), OS (ovarian stroma), V (vessel), CA (collagen arrangement). Yellow asterisk indicates cell presence among ECM fibers. Yellow “x” indicates collagen fiber bundles without cells. Full yellow arrows indicate thick fibers. Dotted arrows indicate thin fibers.

**Figure 6 cells-12-01864-f006:**
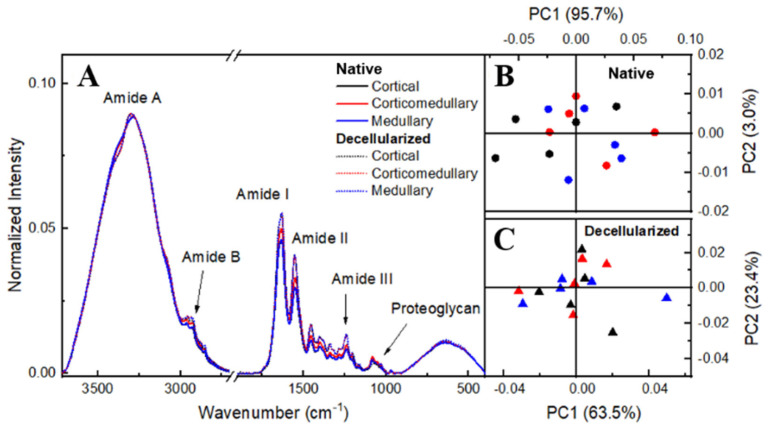
FTIR-ATR spectrum of cortical, corticomedullary, and medullary regions of the porcine ovary (**A**). PCA scores plots of native (**B**) and decellularized (**C**) samples.

**Figure 7 cells-12-01864-f007:**
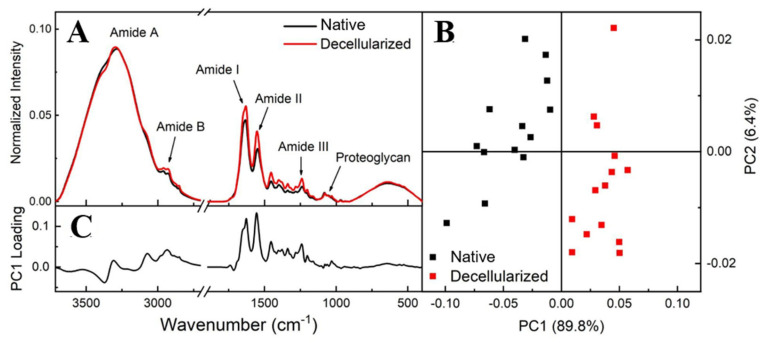
FTIR-ATR spectra from the regions of native and decellularized samples (**A**). PCA score plot comparing native and decellularized samples (**B**). PC1 loading plot (**C**).

**Figure 8 cells-12-01864-f008:**
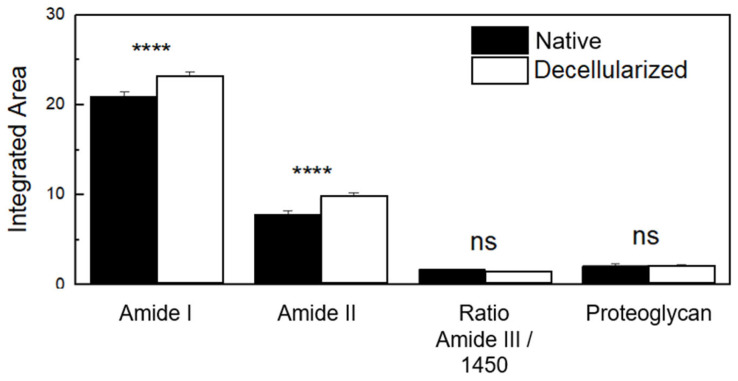
FTIR-ATR integrated band areas of collagen (amide I and amide II) and proteoglycan content and band area ratio of amide III: 1450 cm^−1^ (triple helical structure of collagens) in native porcine ovaries samples and after the decellularization process. ns (not significant); **** *p* < 0.0001.

**Figure 9 cells-12-01864-f009:**
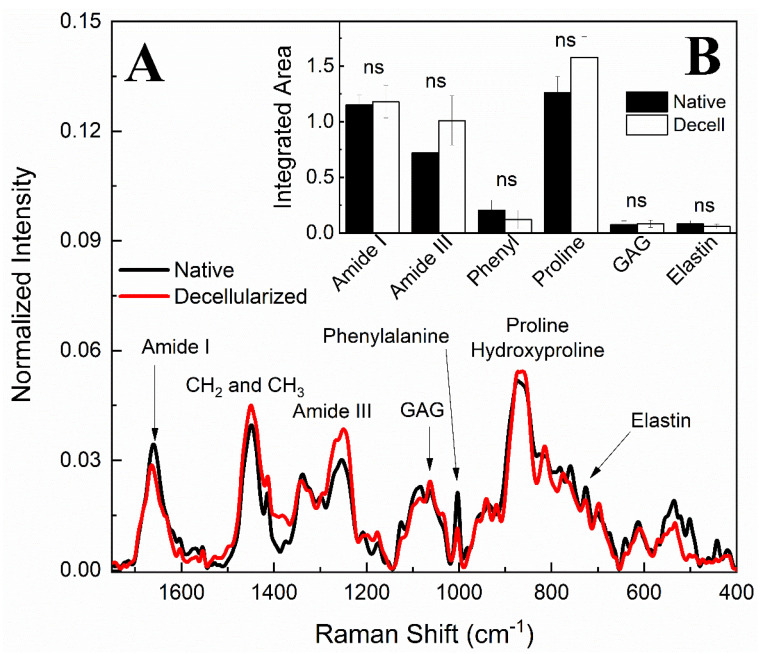
Raman spectra of native and decellularized porcine ovary samples (**A**). Raman bands integrated areas of collagen content (amide I and amide III), phenylalanine (phenyl), proline, hydroxyproline, glycosaminoglycans (GAG), and elastin in native and decellularized (Decell) samples (**B**). ns (not significant).

**Figure 10 cells-12-01864-f010:**
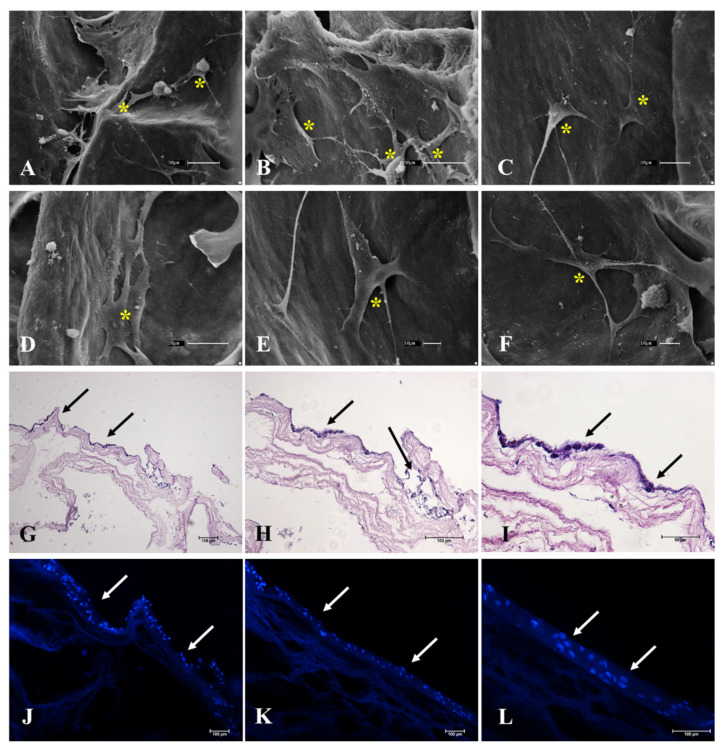
Representative photomicrographs of the adhesion assay performed with 3T3 murine fibroblasts on decellularized ovarian scaffolds after 7 days of culture by SEM microscopy (**A**–**F**), H&E (**G**–**I**), and DAPI staining (**J**–**L**). H&E and DAPI staining shows the cells seeded on the surface of the scaffolds. SEM microscopy highlights the cells attached to the ECM bundles of the scaffold. Yellow asterisk indicates the cells attached to the scaffold. Black and white arrows indicate the cells seeded in the scaffold.

## Data Availability

Not applicable.

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
