# Peer review of "Perfusion and Ultrasonication Produce a Decellularized Porcine Whole-Ovary Scaffold with a Preserved Microarchitecture"

_cells, 2023, doi:10.3390/cells12141864_

Round 1
Reviewer 1 Report
Almeida et al. present a protocol for decellularization of porcine ovaries based on detergent perfusion and ultrasonication to obtain functional whole-ovary scaffolds. Their study is very well written and presented. I look forward to reading future work from this group. I have a few minor comments for this manuscript. Overall, I think this manuscript merits publication.
1. Please provide further info on any post-hoc statistical tests performed for Fig 8 and 9B
2. For section 3.4 - Do you have staining evidence of actin in addition to the DAPI images in 10 J-L? If so, that might be helpful. Also, is there a reason why viability of the attached cells was not quantitatively tested?
Author Response
April 10th, 2023.
Editor-in-Chief
Cells
Dear editor,
Please find enclosed the revised version of our manuscript entitled “Perfusion and Ultrasonication Produces a Decellularized Porcine Whole-Ovary Scaffold with a Preserved Microarchitecture” submitted to Cells.
We addressed all of the reviewer 1’s comments, which were extremely relevant for improving our manuscript. A point-by-point answer is below addressed:
Reviewer #1: Please provide further info on any post-hoc statistical tests performed for Fig 8 and 9B
Answer: We appreciate the suggestions. We added information that is more complete about the statistical analysis. We used Turkey’s post-hoc test. It is highlighted in yellow in Material and Methods section.
Reviewer #1: For section 3.4 - Do you have staining evidence of actin in addition to the DAPI images in 10 J-L? If so, that might be helpful. Also, is there a reason why viability of the attached cells was not quantitatively tested?
Answer: We appreciate the suggestions. Staining for actin as phalloidin is a very good evidence to see the cytoskeleton, but in our assay is not necessary once we saw the cellular membrane extensions in scanning electron microscopy, which demonstrated that cells were able to attach to the ECM. This is possible only by cytoskeleton modulation by the ECM, so only to evidence the nuclei is enough to demonstrate cells presence in the material. Cell viability assay was also proposed, but as we cultured the cells for 7 days and the analysis by SEM and histology demonstrated that cells remained in the material, interacting with it. Therefore, from these findings, we attested that the material supported cell culture.
I hope you find the revised version of our manuscript suitable for publication. Thank you in advance for your consideration.
Sincerely,
Prof. Dr. Ana Claudia Oliveira Carreira
Faculty of Veterinary Medicine and Animal Science
University of São Paulo
Reviewer 2 Report
This manuscript describes the decellularizing porcine ovaries and their characterizations. The authors presented an enhanced decellularization including perfusiong of detergent followed by ultrasonication. Overall, the manuscript is easy to follow and the outcomes are very interesting.
However, the main issue is that most of the results as presented in the manuscript are qualitative. Considering that the sample size (number of ovaries) was five, it is hard to tell whether the results were reproducible from looking at Figures, mostly images. Not sure at this stage whether quantitative analyses (e.g., protein quantification) could be performed but at least image analyses could be performed as indirect comparisons for macromolecules presence and 3T3 cell density. As written, it is hard to tell if the outcomes would be reproducible for all ovaries used in the study.
Author Response
April 10th, 2023.
Editor-in-Chief
Cells
Dear editor,
Please find enclosed the revised version of our manuscript entitled “Perfusion and Ultrasonication Produces a Decellularized Porcine Whole-Ovary Scaffold with a Preserved Microarchitecture” submitted to Cells.
We addressed all of the reviewer 2’s comments, which were extremely relevant for improving our manuscript. A point-by-point answer is below addressed:
Reviewer #2: However, the main issue is that most of the results as presented in the manuscript are qualitative. Considering that the sample size (number of ovaries) was five, it is hard to tell whether the results were reproducible from looking at Figures, mostly images. Not sure at this stage whether quantitative analyses (e.g., protein quantification) could be performed but at least image analyses could be performed as indirect comparisons for macromolecules presence and 3T3 cell density. As written, it is hard to tell if the outcomes would be reproducible for all ovaries used in the study.
Answer: Thank you for the valuable suggestions. We understand that the ECM morphological data is qualitative; however, we chose to use methods more refined to quantify the ECM components, once the morphometry is semi quantitative and may have bias in the quantification. Spectroscopic analysis as FTIR-ATR and Raman are very sensitive methods that detect molecular fingerprints that characterize quantitatively with more reliability than image quantification. The images are representative of the morphological state of the tissue before and after the decellularization. Some papers in the field many times do not show morphological data, only quantifying by biochemical, spectroscopic or proteomic analysis. We chose to put both to demonstrate the presence of the ECM components in the tissue, but the quantification was performed by methods more reliable.
Protein quantification in this stage is not feasible for us, once to quantify every ECM component would be unaffordable for our group. That is why we chose the spectroscopy that is more executable, fast and sensitive than other methods being realistic for our group’s conditions.
Regarding cell density, it is not our purpose to recellularize the tissue, but only demonstrate that the generated material is able to support cell culture. This material is not limited to be applied for ovarian reconstruction, but a very ECM-enriched biomaterial that can be modulated of several ways and for several purposes, in or out reproduction.
I hope you find the revised version of our manuscript suitable for publication. Thank you in advance for your consideration.
Sincerely,
Prof. Dr. Ana Claudia Oliveira Carreira
Faculty of Veterinary Medicine and Animal Science
University of São Paulo
Reviewer 3 Report
In this study, the authors aimed to establish a protocol for decellularization of ovarian tissues, and evaluated the morphology of ECM after decellularization. Using the proposed protocol, the authors showed that cellular components were efficiently removed, with no significant loss of ECM components compared to native tissues, and that the scaffolds exhibited cytocompatibility capable of cell adhesion. Although characterization of the decellularized ovarian tissue has been well studied, the evaluation in recellularization is insufficient, and the superiority of the method in this study to existing methods is still unclear. It is strongly desired to evaluate the functional expression of cells on the decellularized scaffolds using ovary-derived cells or differentiation-induced ovarian cells.
Author Response
April 10th, 2023.
Editor-in-Chief
Cells
Dear editor,
Please find enclosed the revised version of our manuscript entitled “Perfusion and Ultrasonication Produces a Decellularized Porcine Whole-Ovary Scaffold with a Preserved Microarchitecture” submitted to Cells.
We addressed all of the reviewer 3’s comments, which were extremely relevant for improving our manuscript. A point-by-point answer is below addressed:
Reviewer #3: In this study, the authors aimed to establish a protocol for decellularization of ovarian tissues, and evaluated the morphology of ECM after decellularization. Using the proposed protocol, the authors showed that cellular components were efficiently removed, with no significant loss of ECM components compared to native tissues, and that the scaffolds exhibited cytocompatibility capable of cell adhesion. Although characterization of the decellularized ovarian tissue has been well studied, the evaluation in recellularization is insufficient, and the superiority of the method in this study to existing methods is still unclear. It is strongly desired to evaluate the functional expression of cells on the decellularized scaffolds using ovary-derived cells or differentiation-induced ovarian cells.
Answer: We appreciate the comments about the manuscript. At first, it was never our intention to recellularize the generated biomaterial, once the expression ‘recellularized’ or ‘recellularization’ is not used in out manuscript. Our purpose was to generate a whole-ovarian scaffold by a more refined method that preserved with much more efficiency the ECM microarchitecture compared to other methods that takes more time, are more expensive and does not generate a whole-scaffold, which preserved better the ECM and the tissue tridimentionality. We only demonstrated that the material is able to support cell culture and the interaction between the scaffold and cells, not to recellularize it. The main goal of the manuscript is focused in the ECM, because our material is not limited to be used as a scaffold for ovarian reconstruction, but this ECM-enriched biomaterial may have a number of modulations and applications that transcend recellularization. A biomaterial with this level of ECM preservation may be used for several purposed in regenerative medicine and to be translated to for other organs, not being restricted to the ovary, which is the aim of our group, use biomaterial from different organs to unconventional applications, expanding the tissue engineering field. For these reason, we used 3T3 cells, not ovarian cells, to demonstrate that the material supports non-native ovarian cells, highlighting it potential to be applied as a bioscaffold for other tissues and organs.
I hope you find the revised version of our manuscript suitable for publication. Thank you in advance for your consideration.
Sincerely,
Prof. Dr. Ana Claudia Oliveira Carreira
Faculty of Veterinary Medicine and Animal Science
University of São Paulo
Round 2
Reviewer 2 Report
Thank you to the authors for addressing my comments. While I fully understand the limitations and agree with the authors' views, the authors' responses need to be included in the results or discussion section of the manuscript.
Author Response
April 16th, 2023.
Editor-in-Chief
Cells
Dear editor,
Please find enclosed the revised version of our manuscript entitled “Perfusion and Ultrasonication Produces a Decellularized Por-cine Whole-Ovary Scaffold with a Preserved Microarchitecture” submitted to Cells.
We addressed all of the reviewer 2’s comments, which were extremely relevant for improving our manuscript. A point-by-point answer is below addressed:
Reviewer #1: Thank you to the authors for addressing my comments. While I fully understand the limitations and agree with the authors' views, the authors' responses need to be included in the results or discussion section of the manuscript.
Answer: We appreciate the comments made so far. Regarding the reviewer 2’s suggestions, we added to the manuscript and clarified any misunderstandings or justification that we provided to the comments made before. All alterations are highlighted in yellow for your correction.
I hope you find the revised version of our manuscript suitable for publication. Thank you in advance for your consideration.
Sincerely,
Prof. Dr. Ana Claudia Oliveira Carreira
Faculty of Veterinary Medicine and Animal Science
University of São Paulo
Reviewer 3 Report
The explanation of the author's excuses on the purpose of scaffold usage should be added in the manuscript to prevent misunderstandings by readers.
Author Response
April 16th, 2023.
Editor-in-Chief
Cells
Dear editor,
Please find enclosed the revised version of our manuscript entitled “Perfusion and Ultrasonication Produces a Decellularized Por-cine Whole-Ovary Scaffold with a Preserved Microarchitecture” submitted to Cells.
We addressed all of the reviewer 3’s comments, which were extremely relevant for improving our manuscript. A point-by-point answer is below addressed:
Reviewer #1: The explanation of the author's excuses on the purpose of scaffold usage should be added in the manuscript to prevent misunderstandings by readers.
Answer: We appreciate the comments made so far. Regarding the reviewer 3’s suggestions, we added to the manuscript and clarified any misunderstandings or justification that we provided to the comments made before. All alterations are highlighted in yellow for your correction.
I hope you find the revised version of our manuscript suitable for publication. Thank you in advance for your consideration.
Sincerely,
Prof. Dr. Ana Claudia Oliveira Carreira
Faculty of Veterinary Medicine and Animal Science
University of São Paulo